# A novel motorized office chair causes low-amplitude spinal movements and activates trunk muscles: A cross-over trial

**Hendrik Schäfer**[1,2,3]*, **Robin Schäfer**[1,4], **Petra Platen**[1]

1 Department of Sports Medicine and Sports Nutrition, Faculty of Sports Science, Ruhr University Bochum, Bochum, Germany, 2 Department of Rehabilitation Sciences, Faculty of Health, University of Witten/Herdecke, Witten, Germany, 3 DRV Clinic Königsfeld, Center for Medical Rehabilitation, Ennepetal, Germany, 4 Division of Physiotherapy, Department of Applied Health Sciences, University of Applied Sciences, Bochum, Germany

* hendrik.schaefer@rub.de

## Abstract

### Introduction

Inactivity and long periods of sitting are common in our society, even though they pose a health risk. Dynamic sitting is recommended to reduce this risk. The purpose of this study was to investigate the effect of continuous passive motion (CPM) conducted by a novel motorized office chair on lumbar lordosis and trunk muscle activation, oxygen uptake and attentional control.

### Study design

Randomized, single-session, crossover with two periods/conditions.

### Methods

Twenty office workers (50% women) sat for one hour on the motorized chair, one half with CPM, the other not. The starting condition (CPM/no CPM) was switched in half of the sample. The participants were equipped with a spirometric cart, surface EMG, the Epionics SPINE system and performed a computer-based test for attentional control (AX-CPT). Outcomes were lumbar sagittal movements and posture, number of trunk muscle activations, attentional control and energy expenditure.

### Results

The CPM of the chair causes frequent low-amplitude changes in lumbar lordosis angle (moved: 498 ± 133 vs. static: 45 ± 38) and a higher number of muscle activations. A periodic movement pattern of the lumbar spine according to the movement of the chair was observed in every participant, although, sitting behavior varied highly between individuals. Attentional control was not altered in the moved condition (p = .495; d = .16). Further, oxygen uptake did not increase higher than 1.5 MET.

**Data Availability Statement:** All data (including raw data) and study documents are available on OSF: https://osf.io/7szrq/.

**Funding:** PP received financial support from the European Commission (https://ec.europa.eu) via

the European Regional Development Fund (ERDF) in the form of a grant (0400298). No additional external funding was received for this study. The funders had no role in study design, data collection and analysis, decision to publish, or preparation of the manuscript.

**Competing interests:** The authors have read the journal's policy and have the following competing interests: The chair in this study was a prototype developed and provided by the start-up company Vintus (https://vintus.de/en/). The company had no role in study design and analysis, decision to publish, or preparation of the manuscript. There are no additional patents, products in development or marketed products associated with this research to declare. This does not alter our adherence to PLOS ONE policies on sharing data and materials.

## Conclusion

The effects of the motorized chair can be particularly useful for people with static sitting behavior. Further studies should investigate, whether CPM provides the assumed beneficial effects of dynamic sitting on the spine.

## Introduction

„Is sitting a lethal activity?"–This title comes from the New York Times Magazine [1]. The article refers to a study by the American Cancer Society [2] which highlights a higher mortality risk for people with long periods of sitting. According to Patel, this risk increases even further when the long periods of sitting are accompanied by general inactivity. Nowadays a sedentary lifestyle is omnipresent. An increasing number of people spend long periods in a sitting position, both at work and during leisure time [3]. A large proportion of them has a sedentary occupation. In some service sectors, 77% to 86% of employees work at a computer [4]. A descriptive study of more than 35,000 French workers shows that the average sitting time is 11.3 hours per day. Almost half of this is attributed to driving and working time [3].

Social structures are making people increasingly physically inactive. The Sedentary Behaviour Research Network defined this as follows: "any waking behavior characterized by an energy expenditure $\leq$ 1.5 METs while in a sitting or reclining posture" [5]. The metabolic equivalent of task (MET) is defined as task energy expenditure divided by resting energy expenditure (3.5 ml per kilogram body weight and minute) [6]. Physical inactivity is increasingly perceived as a health risk for the global society. Sedentary work is associated with a higher risk of type-2-diabetes, cardiovascular disease, cancer, and general and cardiovascular mortality [7–9]. Explanations for the increased risk can be derived from activity restriction studies [10]: Prolonged inactivity negatively affects lipoprotein lipase activity [11], glucose tolerance [12], and cardiac stroke volume and output [13].

In addition, a higher prevalence of back pain among workers with computer workstations has been reported [14, 15]. Although it would be expected that long periods of sitting are associated with the occurrence of back pain, there is no clear position in science [16, 17]. It is postulated that sitting for more than seven hours per day leads to an increased risk of back pain [18]. In contrast, no clear correlation between sitting times and the occurrence of back pain could be found [19, 20]. Back pain is described as a highly multifactorial condition, which may explain the lack of link [21, 22]. Therefore, sitting does not appear to be a clear risk factor. Furthermore, the preferred sitting posture, which is maintained over a long period, seems to influence the development of musculoskeletal complaints [23]. In the past, sitting upright and static sitting behavior were considered optimal [24]. However, this recommendation has now been replaced by dynamic sitting [25]. A sitting position should not be described as the best, because there seems to be no ideal position. In addition, the individually preferred lumbar lordosis when sitting is considered preventive [26]. Frequent changes in sitting position lead to variable loads on the musculoskeletal system and the spinal column geometry [27]. A dynamic sitting behavior can vary the loading conditions of the spinal segments, which induces an effective pumping mechanism in the intervertebral discs. It is believed that this mechanism is crucial for the nutrition of the intervertebral discs and for reducing degenerative changes [28].

The rethinking and recommendation of dynamic sitting have an impact on the development and design of office chairs [29]. In recent years, a variety of different ergonomic and movable office chairs have been developed to support dynamic sitting [30]. However, there are no motorized chairs with continuous passive motion (CPM) on the market. Although CPM in

sitting was first investigated by Reinecke and Hazard [31]. A few years later van Deursen et al. and Lengsfeld et al. showed with various studies positive effects of a passive moved sitting, such as increased muscle activity, reduced spinal shrinkage or pain [32–39]. However, a randomized multicenter study by Lengsfeld [40] could not show any positive effects of CPM on employees with (low) back pain (LBP). The less passive movement chosen in this study was criticized by Kumar [41]. He argued that axial rotation of 1.6˚ has no effect. The *Vintus* project funded by the European Union follows this approach to develop a motorized office chair. A prototype allows a horizontal movement of the seat pan. This supports dynamic sitting due to a continuous passive movement that mobilizes the seated person. Regarding the criticism of Kumar [41], translatory movement with greater deflection was chosen for the motorized chair. Except for the work of Lengsfeld and van Deursen, the passively moved sitting is scarcely investigated. Furthermore, the effects of dynamic sitting on the body are unclear. O'Sullivan et al. [42] postulated that dynamic sitting may increase the total amount of spinal motion. However, they emphasized in their systematic review the limiting or conflicting evidence that dynamic sitting reduces spinal shrinkage or alters lumbar posture or trunk muscle activation. The motorized chair was examined under the assumption that CPM causes positive effects. But it is unclear how the chosen passive movement affects spinal posture and muscle activity. Additionally, little is known of the effects of CPM ont the ability to concentrate on tasks.

Therefore, the study aimed to investigate the effects of CPM in sitting on lumbar lordosis and trunk muscle activation. In addition, the metabolic rate was examined to assess whether the moved sitting leads to increased MET $\geq 1.5$ and a test for the attentional control was conducted, to account for a potential distraction of the participants.

## Methods

### Study design

A single session, repeated measures, crossover design was chosen for the laboratory study. Each participant sat 60 minutes on the motorized chair, 30 minutes under moved (MC) and 30 minutes under static conditions (SC). The outcomes were attentional control, oxygen uptake, trunk muscle activity, and lumbar lordosis. The order of the conditions (MC/SC) was allocated by the investigators to achieve balance in group size and gender. The study was approved by the local Ethics Committee of the Faculty of Sport Science of Ruhr University Bochum (EKS V 08/2020) and was conducted in accordance with the Declaration of Helsinki. Participants received verbal and written information on the procedure and purpose of this study, as well as on possible risks. Consequently, written informed consent was retrieved from all participants. All had to declare that they were free of any health restrictions, were able to work and to follow the strict precautions of Covid-19.

The outcomes were measured via a computer-based test for attentional control, a stationary spirometric system, sEMG and the Epionics SPINE system. After the placement of the electromyography (sEMG) electrodes, maximal voluntary contractions were performed. During a 10-minute familiarization with the chair, the subjects completed the Chronic Pain Grade (CPG) and International Physical Activity Questionnaire (IPAQ). The computer-based continuous performance test was started after the attachment of the spirometry and the Epionics SPINE system. The study was conducted in a typical office environment, where the computer workstation was aligned according to ergonomic recommendations [43]. The knee and hip angles were set to 90˚. Smaller subjects used a footrest. The seat was fixed in a horizontal plane and the backrest was aligned orthogonally. The table was not adjustable, but all subjects had a nearly 90˚ elbow angle. The participants were alone in the room during the test but were

filmed from behind. The investigator was present for less than 5 minutes in the beginning, in the middle, and at the end of the test.

## Participants

Ten women and ten men were recruited (M ± SD: age = 26.6 ± 4 years; height = 176.3 ± 7.8 cm; body mass = 72 ± 12.1 kg). Due to Covid-19 restrictions, only employees and students from the Faculty of Sport Science of Ruhr University Bochum could participate in the study. The mean daily sitting time of the recruited group was 7.8 ± 2.7 hours. Physical activity level was assessed with the validated IPAQ [44, 45]. Regarding the IPAQ guidelines [46], four subjects were excluded because of incomplete data. The MET-minutes/week for walking, moderate, and vigorous activities were 1335 ± 1138, 1148 ± 1146, and 3240 ± 2551, respectively (n = 16). The validated CPG questionnaire [47] showed for all subjects almost no pain-related disability (disability score: 6 ± 6.9%) and low pain intensity (characteristic pain intensity (CPI): 21.8 ± 16.4%). Two subjects had higher pain intensity levels than 50%.

## Motorized chair

The developer used a classic office chair without armrests for their prototype (Fig 1a). The five-legged movable pedestal is fitted with a gas pressure spring, which allows the seat height to be adjusted. The backrest position is variable but has been set to an angle of 90°. Two electric motors, mechanics, and a rechargeable battery are placed underneath the movable seat. Garner & Rigby [48] showed a pelvic motion pattern during walking that corresponds to the figure of eight. Therefore, the movement profile of an eight was chosen for the motorized chair to

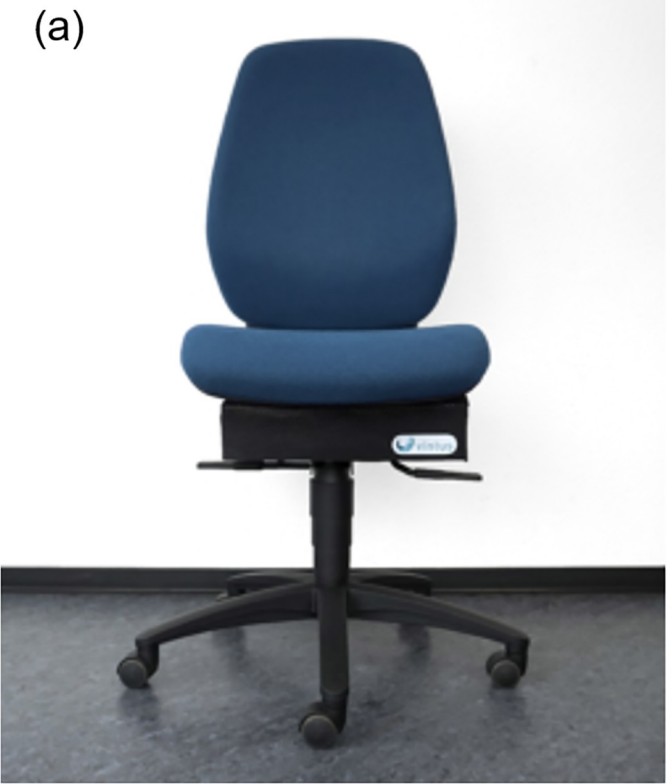

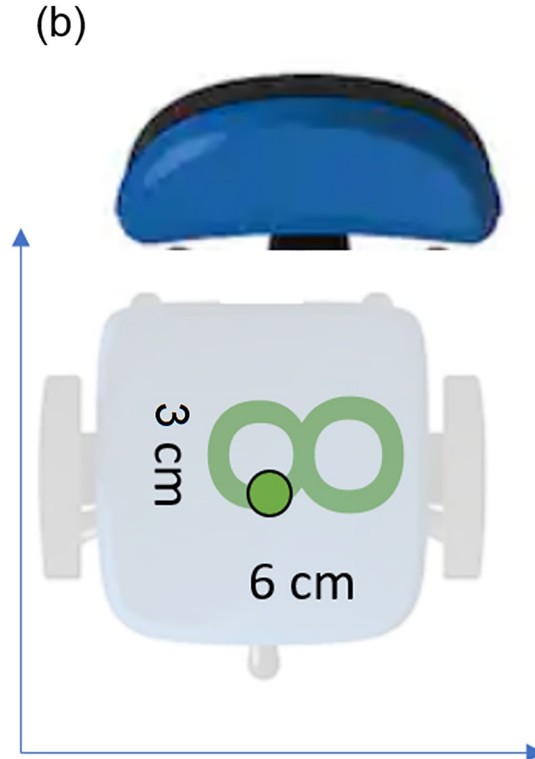

**Fig 1. The motorized office chair.** (a) prototype (b) movement profile of an eight and the distortion of 6 x 3 cm.

imitate walking and distortion was set to 6 cm in X and 3 cm in the Y direction (Fig 1b). The speed of movement was set to 56 mm/s, which is the maximal velocity of the prototype.

## Attentional control

The modified version of the continuous performance test (AX-CPT) [49] was used to assess the attentional control (Fig 2). The test lasted 60 minutes; 30 minutes for each half were included in the analysis.

The AX-CPT is a computer-based test. Small letters appear continuously on a black screen. Each letter is displayed for 300 ms, after a pause of 1000 ms the next letter appears for 300 ms. The letters are displayed on a cue-probe basis, i.e. five consecutive letters form a series. The first letter appears red and forms the cue, the last one is also red and means probe. The letters in between are white and have no meaning. The participant was instructed to remember the cue until the fifth letter (probe) appears. Whenever the cue "A" followed by probe "X" appears, the answer should be "Yes". For any other possible cue-probe combination, the participant was asked to press the "No" button. Mistakes were indicated by an acoustic signal. For the evaluation, a quotient was formed from the number of cue-probe series and the correctly detected cue-probe series, which is expressed as a percentage. One advantage of the test is the standardization of the task.

## Oxygen uptake

The spirometric measurement was performed with an Oxycon Pro (Carefusion Netherlands, Houten, Netherlands). Consistency and good reliability of measurements were shown [51, 52].

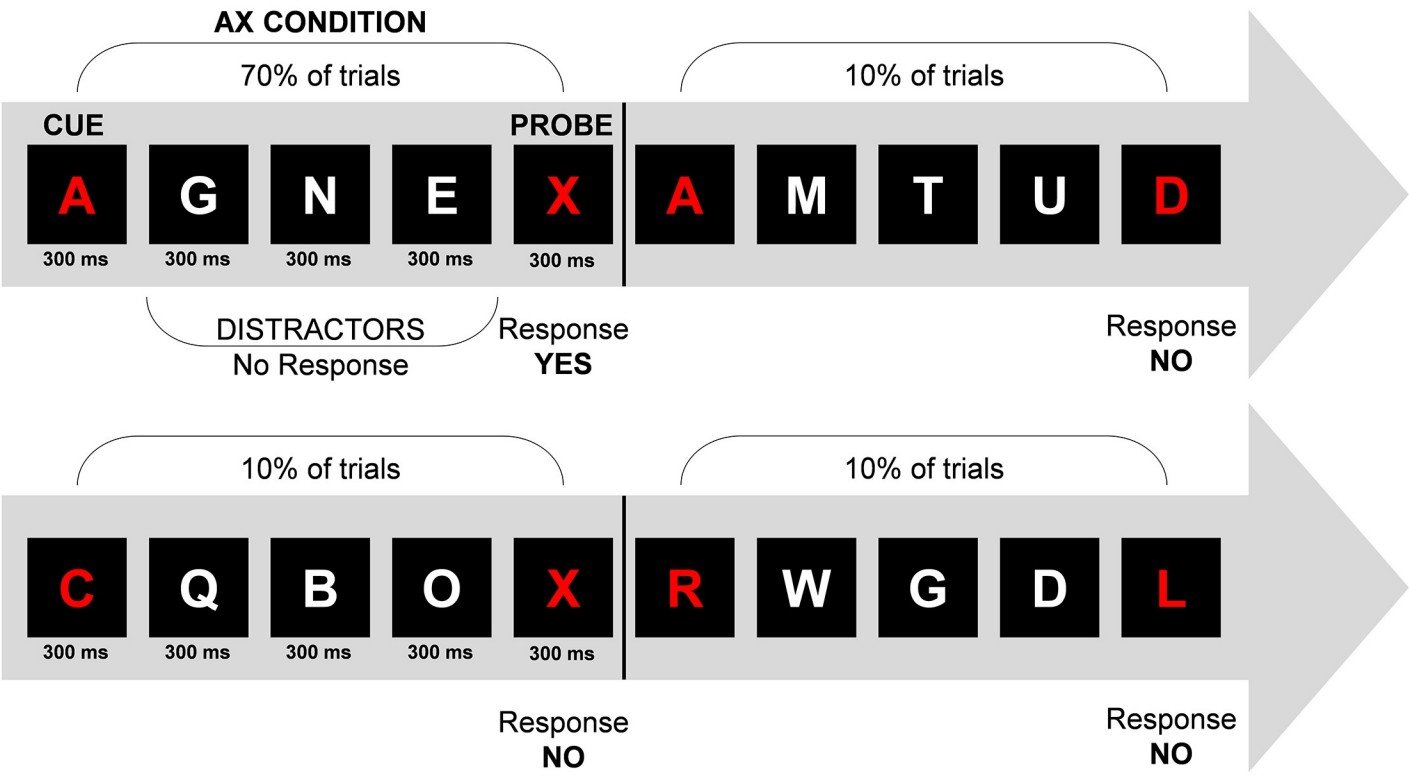

**Fig 2. Continuous performance test.** Examples of series of the AX-CPT and their distribution ([50]; modified).

The device was calibrated before the test. The face mask with a filter was attached airtight. The measurement was performed at room temperature between 20°C and 22°C in an air-conditioned laboratory. Breath-by-breath $\dot{V}O_2$, $\dot{V}CO_2$, and the respiratory minute volume (*VE*) were recorded. $\dot{V}O_2$ data were normalized to the bodyweight. Breath-by-breath data were smoothened to 5 second intervals before analyzing the data. Energy expenditure (EE) in kilocalories was calculated by multiplying the $\dot{V}O_2$ by the factor 4.82 [53].

## Trunk muscle activity

To assess trunk muscle activity, wireless surface electromyography (sEMG) (Aktos; Myon, Schwarzenberg, CH) was used. The sampling frequency was 2000 Hz. It has been shown that the Myon system is a reliable measurement (interclass correlation coefficient, ICC > 0.80; intra-session and inter-day testing) [54]. The set-up of the sEMG was in line with the SENIAM standards [55]. The skin was shaved, prepared with an abrasive skin gel, cleaned, and disinfected with an alcoholic solution to reduce skin impedance. The bipolar, circular (Ø 24 mm) AG/AgCl-Sensor electrodes (Kendall, H124SG, Hydrogel; Covidien Ltd, IRL) were placed with a standardized inter electrode distance of 20 mm over the muscle belly in the direction parallel to the muscle fibers. Both left and right *lumbar m. erector spinae (LES)* and *m. rectus abdominus (REC)* were chosen for the sEMG. Electrodes for LES were placed 2 cm on the side of the processus spinosus at heights L4 and L3. The placement for REC was 1 cm above the umbilicus and 2 cm lateral to the midline. Adequate placement and plausible data sampling were immediately proven by checking the sEMG output.

Maximum voluntary isometric contractions (MVIC) were performed in a standardized setting using a dynamometer (Back Module, Isomed 2000; D&R Fertsl GmbH, GER). The participants sat with fixed legs, hips, shoulders, and a flexed upper body to an angle of 60°. A short warm-up with three trials at 30%, two trials at 50%, and one trial with 70% of subjectively estimated maximal contraction was performed. Then, participants were asked to push as hard as possible against the resistance for three seconds. Two MVIC trials were recorded with a 1-minute break in between.

The sEMG data were processed by a 500 Hz low-pass and 30 Hz high-pass fourth-order Butterworth filter. The 30 Hz high-pass corner frequency was utilized to remove movement and ECG artifacts [56]. Root mean square amplitudes were calculated by moving a 250 ms sliding window over the data. Movement artifacts were removed by screening the data. Test data were normalized to the maximum value from the MVIC. Data are shown as a percentage of MVIC.

## Spinal posture

Lumbar spinal posture was assessed with the Epionics SPINE system (Epionics Medical GmbH, Potsdam, GER). This system consists of two flexible strain gauge-based sensor tapes with twelve 2.5 cm segments and triaxial accelerometers at the ends of each tape. The strips were placed within two hollow patches on the back 7.5 cm paravertebrally to the posterior midline, i.e. the line of the spinous processes, starting at the first sacral vertebra. The sensor strips were connected to a memory unit (size: 12.5 cm by 5.5 cm, mass: 80 g). The data were sampled at a frequency of 50 Hz. The system measures the curvature of the back and converts it into angle degrees. The stripes were only aligned with one standardized anatomical point. Hence no conclusions can be drawn about the angles of the individual vertebra segments. However, lordosis and kyphosis and their transition can be assessed. Therefore, data from the two sensor stripes were averaged. Calculated angles were summed up to one lumbar lordosis angle. For the analysis, movements of the lumbar spine in the sagittal plane up to the

lumbothoracic transition were used. The data output was generated with MATLAB routines developed by the manufacturer. Reliability and accuracy (ICC >.98) were shown in previous studies [57–61].

## Data processing

To detect the number of muscle activations and movements and their magnitude, the sEMG and Epionics data were processed as follows: the data were smoothed with a slightly moving average, the local minimum and maximum were searched for, each local minimum was subtracted from the following maximum, and this maximum was subtracted from the following minimum. This resulted in positive and negative peak-to-peak amplitude. Amplitudes greater than 1° were considered for further analysis. The arithmetic mean of the positive and absolute negative peak-to-peak amplitude was defined as one activation and one movement. This definition was chosen because activation includes the on- and offset and the repetitive movement of the chair leads to similar positive and negative angle changes. This definition is also in line with previous research [62, 63]. To estimate the individual total movement in each condition, the number of movements was multiplied by the average movement amplitude. The sEMG and Epionics data were synchronized to assess the spinal movements and their corresponding muscle activity. Therefore, one sEMG sensor was attached to one Epionics accelerometer. By tapping the sensors twice, peaks are generated that allow data synchronization.

## Statistical analysis

Data are shown as arithmetic mean (M) ± standard deviation (SD). The mid 20 minutes of all measurements, with exception of the AX-CPT, were used for the statistical analysis. Hence, time intervals are from minute 5 to 25 and minute 35 to 55. The intervals were chosen because the investigator was present at the start and the end of each half. All data processing steps were performed with MATLAB (vR2019b, MathWorks Inc., Natick, USA). Figures were created with OriginPro, (v2020b, OriginLab Corporation, Northampton, MA, USA) and contain Box-Whisker Plots, where the Whiskers cover the data up to 1.5 of the interquartile range.

The statistical analysis is in line with the recommendations of Wellek & Blettner [64]. The values of the first and second half were summed and subtracted for each participant. Unpaired t-tests were performed with the starting condition as the independent variable. The p-value of the sum was used to assess carryover effects, whereas the p-value of the difference was used to detect significant changes. Sensitivity analysis by removing outliers was performed if such were present. Normal distribution was checked using the Shapiro-Wilk Test. Non-normally distributed data were analyzed unchanged [65]. All statistical analyses were determined using IBM SPSS Statistics (v22, SPSS Inc., Chicago, USA) and statistical significance was defined at p ≤.05.

## Results

### Attentional control

We found no significant differences in attentional control between moved and static sitting (MC: 96.4 ± 3%; SC: 96 ± 3%; p = .495; d = -.155). However, the data of the attentional control variable was not normally distributed. No carryover-effects were detected (p = .744). Removing an outlier (Fig 3a) led to slightly higher, but still non-significant, effects for the moved condition (MC: 97 ± 2%; SC: 96 ± 3%; p = .059; d = -.389; n = 19). The movement of the chair did not alter the attentional control.

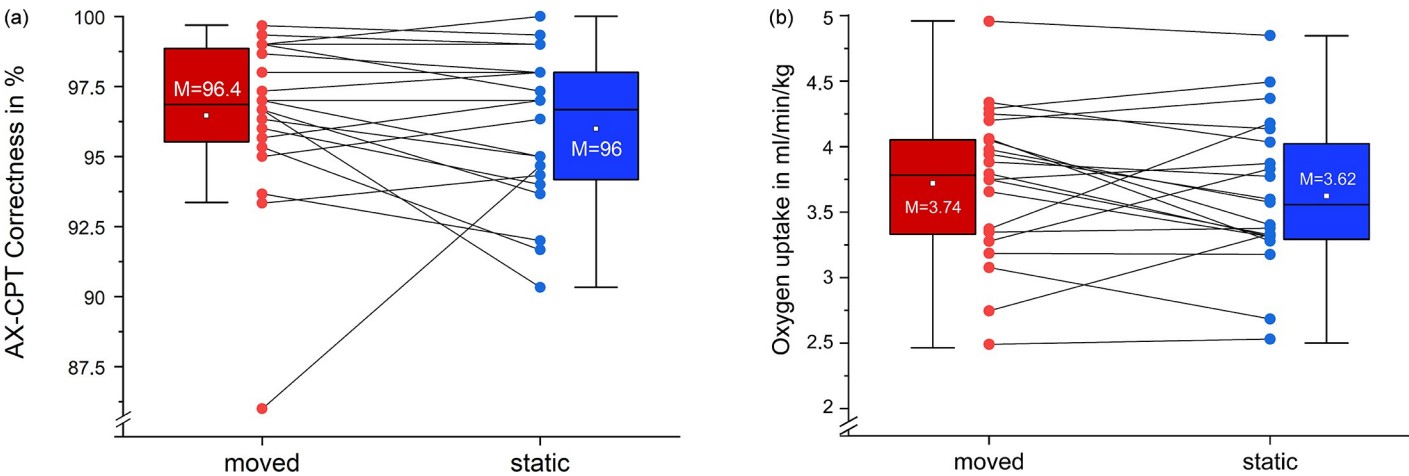

**Fig 3.** Results of the continuous performance test (a) and oxygen uptake (b). (a) Correct answers in percentage for the moved condition (red) and the static condition (blue) (N = 20). Removing an outlier (1) led to a difference among conditions of 1% (p = .059). (b) Oxygen uptake for the moved condition (red) and the static condition (blue) (N = 20).

## Oxygen uptake

We found no significant differences in the mean oxygen uptake between moved and static sitting (MC: 3.74 ± .59 ml/min/kg; SC: 3.62 ± .59 ml/min/kg; p = .119; d = -.213) (Fig 3b). The mean metabolic equivalent of task was 1.07 MET (± .17) for the moved condition and 1.03 MET (± .17) for the static condition. No carryover-effects were detected (p = .734). The mean oxygen uptake for the first half was lower compared to the second half (FH: 3.58 ± .6 ml/min/kg; SH: 3.78 ± .58 ml/min/kg; p = .022). Seven subjects had higher $\dot{V}O_2$ -values under the static condition.

## Trunk muscle activity

While muscle activity of the *m. rectus abdominus* was barely detectable (l. REC + r. REC: MC: .3% ± .2%; SC: .3% ± .2%; p >.05), we only investigated activity of the lumbar *m. erector spinae* (LES). Further, one participant (ID16) from the MC/SC group was excluded because of implausible MVIC data, yielding 19 participants for this analysis.

Between moved and static sitting, we found a significant difference in the number of movements, but not in the mean activation and the mean amplitude (Table 1). Data inconsistencies occurred quite regularly (e.g., large differences between left and right side, low signal), but we did not exclude these data. Normality assumptions were violated for the right LES mean muscle activity and the number of activations for left and right LES.

Fig 4a shows a representative activation pattern of the left LES for one movement cycle. The peak activation (~8 MVIC%) is followed by a period with less activity, a second smaller peak, and ends with almost no activity. Peak activation occurred at posterior seat positions.

## Spinal posture

Data from all 20 participants were included in the analysis. The mean lumbar lordosis angle was highly variable for MC 0.3° ± 14.9° and for SC 2.8° ± 15.4° (p = .058; d = .165). Three examples show these variations and also different sitting behavior (Fig 5). Participant ID01 sat stable in the first half with CPM and moved frequently in the static condition; Participant ID04 sat similarly stable in both conditions with almost no greater movements; Participant

**Table 1. Participant's demographics.** Participants demographics. Sitting time is derived from International Physical Activity Questionnaire (IPAQ) item 7. Data from 8 participants each were used for the IPAQ. The disability score (DIS) and characteristic pain intensity (CPI) are components of the Chronic Pain Grade (CPG) questionnaire.

| | | Female = 10 | Male = 10 |
|---|---|---|---|
| age [yrs] | | 24.5 ± 3.6 | 28.6 ± 3.3 |
| hight [cm] | | 170.4 ± 5.3 | 182.1 ± 5.0 |
| weight [kg] | | 63.4 ± 7.6 | 80.5 ± 9.5 |
| sitting time [h] | | 8.9 ± 2.9 | 6.7 ± 2.0 |
| CPG [%] | DIS | 7.7 ± 7.0 | 4.3 ± 6.9 |
| | CPI | 25.3 ± 15.4 | 18.3 ± 17.4 |
| IPAQ [MET-minutes/ week] | vigorous | 3800 ± 3435 | 2680 ± 1193 |
| | moderate | 1320 ± 1478 | 975 ± 749 |
| | walking | 1961 ± 1276 | 710 ± 502 |
| | total | 7081 ± 5568 | 4365 ± 1579 |

ID20 showed many movements with a great range of motion and a repetitive pattern in both conditions, i.e. short extension movements are followed by long-lasting periods of flexion. We observed similar patterns in the other participants.

The cyclic angle change (Fig 5a) caused by the movement of the chair can be found in all participants. In Fig 4, muscle activity and the lordosis (vertical axis) are shown as functions of

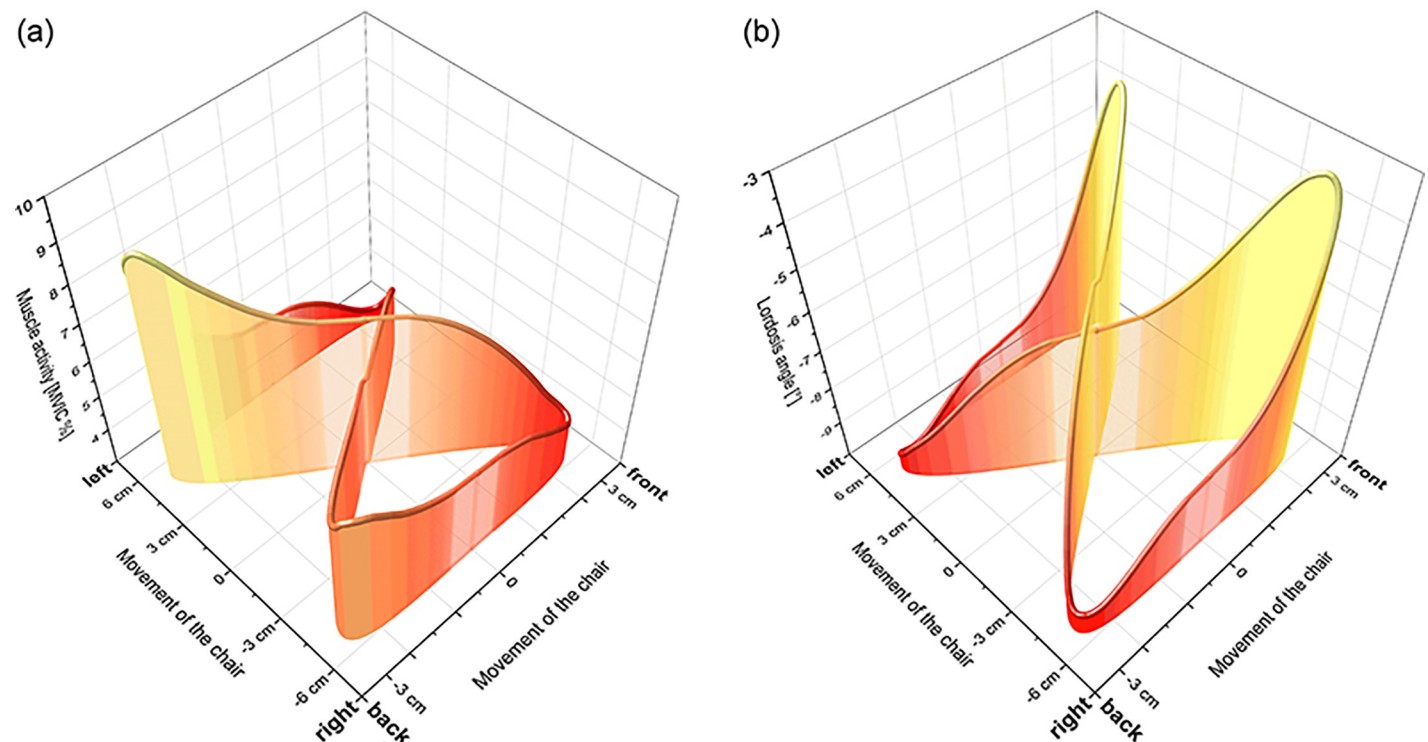

**Fig 4. 3-D visualization of trunk muscle activity and lumbar lordosis (ID03).** The seat movement profile "eight" causes changes in the muscle activity of the left *m. erector spinae* (a) and the lumbar lordosis (b). The x- and y-axis refer to the movement of the chair, whereas the z-axis shows the respective parameter. High values are shown in light yellow and low values are in dark red. This example is derived from subject ID03. Epionics and EMG data were synchronized hence the figure shows the same time frame.

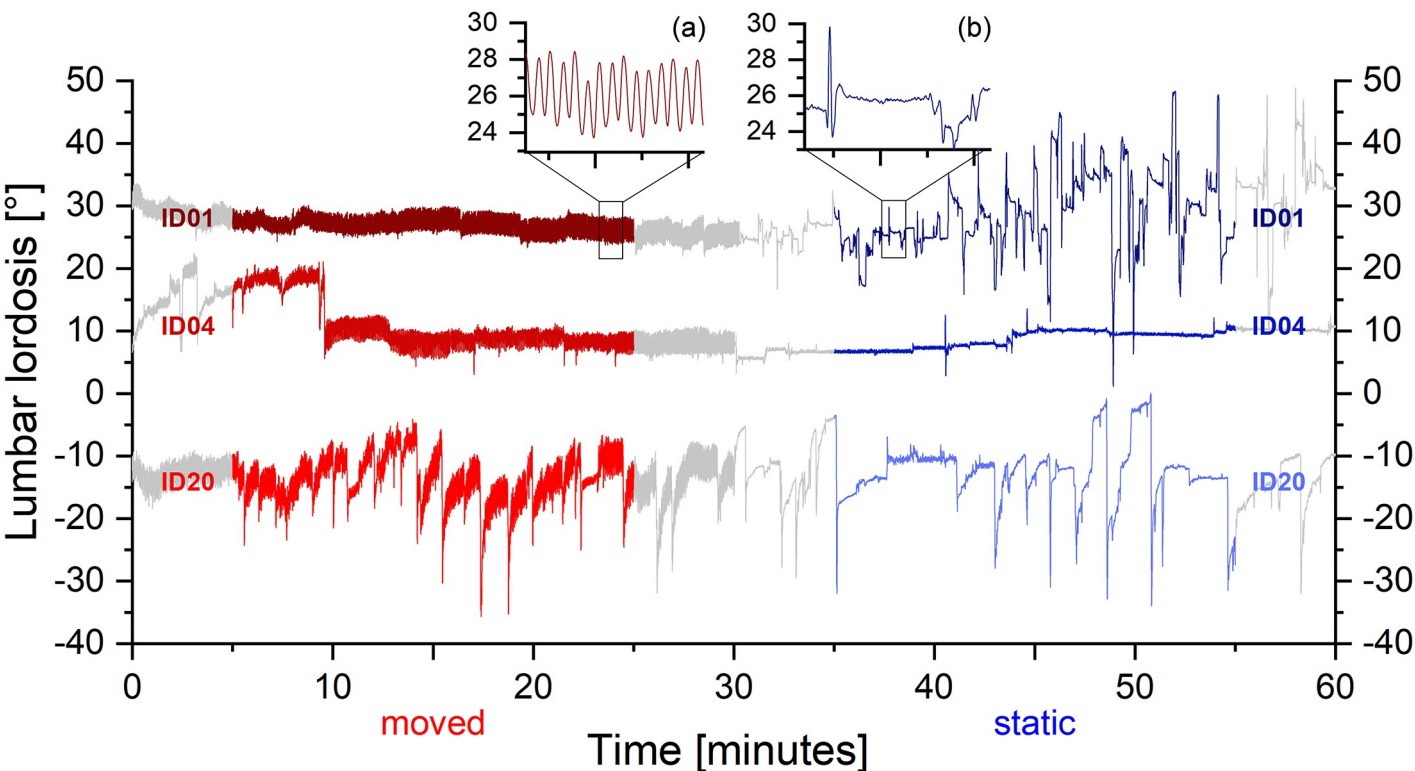

**Fig 5. Test data of lumbar lordosis changes (ID01, ID04, ID20).** The analyzed 20 minutes of the moved first half is shown in red and the static second half is shown in blue. The test lasted 60 minutes and not analyzed data are shown in grey. Subjects had different mean lordosis angles. Magnified windows show the cyclic angle change of the moved sitting compared to the static sitting.

the movement pattern of the chair ("8" in the horizontal plane) for one participant. Lumbar flexion occurs when the seat pan moves forward and extension when backward. Thus, the changes in both parameters were highly dependent on the seat pattern movement. The range of motion in this example was 5˚.

Between moved and static sitting, we found a significant difference in the number of movements and total angle change, but not in the mean lumbar lordosis and the mean angle change (Tables 2 and 3). No carryover effects were found for these variables.

Movement patterns were highly variable between individuals. Two subjects had a higher mean amplitude under MC. Three subjects showed no movements under SC. 15 participants had higher mean amplitude and greater deviations for SC compared to MC. The three subjects

**Table 2. Muscle activity data.** Mean muscle activity, mean activity amplitude, and the number of activations for left and right lumbar erector spinae (LES) (n = 19).

|  | **Moved** | **Static** | **p-value** | **Cohen's d** |
|---|---|---|---|---|
| r. LES mean [MVIC %] | **2.4 ± 2.0** | **2.6 ± 2.8** | p = .551 | d = .098 |
| l. LES mean [MVIC %] | **2.9 ± 4.4** | **2.0 ± 3.5** | p = .402 | d = -.141 |
| r. LES mean amplitude [MVIC %] | **2.7 ± 2.0** | **3.1 ± 2.8** | p = .295 | d = .156 |
| l. LES mean amplitude [MVIC %] | **3.0 ± 1.6** | **3.2 ± 2.0** | p = .609 | d = .117 |
| r. LES number of activations | **205 ± 187** | **53 ± 70** | p = .001 | d = -1.074 |
| l. LES number of activations | **200 ± 200** | **39 ± 53** | p = .002 | d = -1.097 |

**Table 3. Lumbar lordosis data.** Arithmetic mean of the Epionics measurement for both conditions. Variables are mean lumbar lordosis angle, mean angle change, the number of movements and total movement (N = 20).

|  | Moved | Static | Significance | Cohen's d |
|---|---|---|---|---|
| Mean lumbar lordosis [°] | 0.3 ± 14.9 | 2.8 ± 15.4 | p = .058 | d = .165 |
| Mean angle change [°] | 3.4 ± 1.4 | 4.3 ± 3.5 | p = .275 | d = .359 |
| Number of angle changes | 498 ± 133 | 45 ± 38 | p < .001 | d = - 4.615 |
| Total angle change [°] | 1649 ± 698 | 285 ± 422 | p < .001 | d = - 2.366 |

with the highest mean amplitude for SC were ID10, ID12, and ID14 (Fig 6a). The number of movements was higher for MC with a huge effect size (Fig 6b). The high mean amplitude of ID10, ID12, and ID14 and the average number of movements led to divergent high total angle change (Fig 6c). However, the average total movement was higher with CPM.

## Discussion

This study gives first insights into the effects of a novel motorized office chair with CPM on lumbar lordosis and trunk muscle activity. We found a periodic movement of the lumbar spine according to the movement of the chair in all participants besides highly varying sitting behavior. Attentional control was not altered in the moved condition. Further, oxygen uptake did not increase higher than 1.5 MET, which is a supposed threshold for sedentary behavior.

### Attentional control

The continuous performance test was a useful tool to ensure a standardized measurement. The continuous task guaranteed that the subjects were not distracted. Lengsfeld et al. [40] chose a very small movement of the seat for their study. They hypothesized that a larger dislocation could affect the ability to concentrate. This assumption seems logical, as the attentional demands increase from sitting to standing to walking [66]. But this could not be confirmed by the studies of Henz et al. [67, 68]. They reported an improvement in mathematical performance during dynamic sitting [67]. In addition, dynamic work which includes dynamic sitting and standing led to better attentional and vigilance performance [68]. Our findings showed similar results in attentional control for both conditions. After the removal of an outlier, the

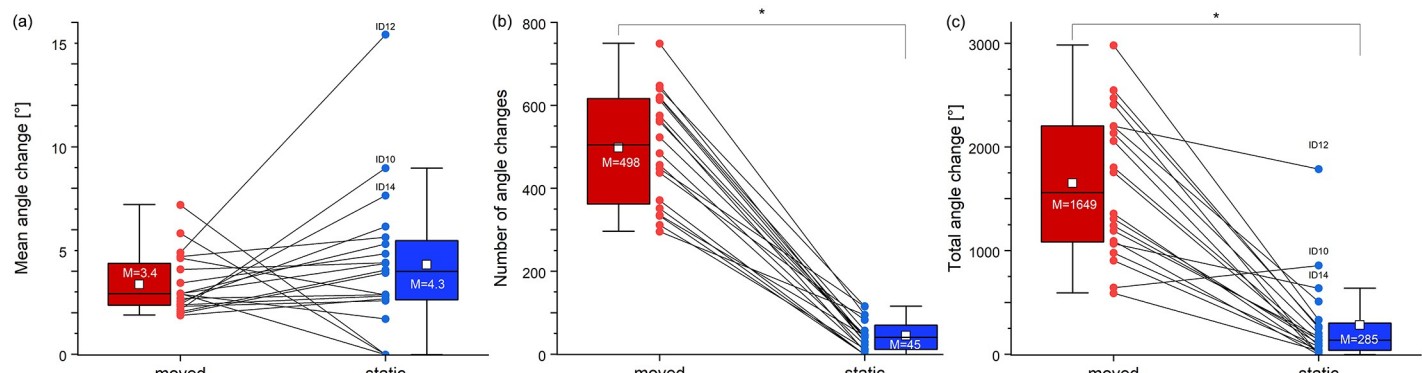

**Fig 6. Mean lumbar lordosis changes, the number of movements and the total movement.** Box-Whisker plots of the Epionics measurement for both conditions (moved: red; static: blue). (N = 20): (a) mean lumbar lordosis angle change; (b) number of angle changes; (c) total angle changes in degrees, which is calculated by multiplying the number of changes and the mean angle change.

average performance for MC was 1% higher(p = .059). Henz et al. [68] observed longer durations of up to 4 hours, while the AX-CPT used in this study lasted 60 minutes. Potential improvements could be found by prolonged use of the motorized chair in this study. Even if we could not show any improvements there seems to be no evidence for Lengsfeld's hypothesis. However, the effect of CPM on attentional control should be further investigated.

## Oxygen uptake

*"Oxygen uptake ($\dot{V}O_2$) represents oxygen consumption by tissues and is a function of cardiac output and the difference in oxygen content between venous and arterial blood"*

- M. Gedeon [69].

The motorized chair causes passive movement of the seated person. Whether this leads to an increased oxygen demand of the tissues was evaluated by measuring the oxygen uptake. The mean MET was 1.07 (± .17) for MC, which is in line with the definition of sedentary behavior. Our obtained mean value for SC (1.03 MET) is similar to the result (1.02 MET) of Synnott et al. [70]. They investigated dynamic sitting and found a mean MET value of 1.20. The difference between dynamic sitting and moved sitting could be explained by lower muscle activation due to passive movement. The calculated energy expenditure (EE) per hour in kilocalories was 77.9 kcal/h (326.5 kJ/h) for MC and 75.4 kcal/h (315.9 kJ/h) for SC. The difference of 2.5 kcal/h (10.6 kJ/h) is just a small effect (d = -.213). Gibbs et al. [71] investigated EE of sitting (70.8 kcal/h), standing (79.1 kcal/h), and a combination of them (76.3 kcal/h). Standing led to 8.3 kcal/h higher EE compared to sitting. This is in line with a recent meta-analysis by Saeidifard et al. [72]. Accordingly, the difference between sitting and standing is 9 kcal/h. They describe the higher EE as modest and consider long-term intervention (most days of a year) to be potentially effective. We could not detect any statistically significant differences between MC and SC. Assuming a positive effect, it would be too small to be of practical relevance. However, CPM could have a meaningful effect on the individual level. Seven subjects with higher $\dot{V}O_2$-values under SC had mostly high total angle changes, which could explain the difference in EE (e.g. Fig 6c; ID10, ID12). People with variable sitting habits (e.g. Fig 5; ID01) may not benefit from CPM in the same way as those with static behavior (e.g. Fig 5; ID04).

## Trunk muscle activity

The main difference between dynamic chairs and the chair in this study is the motorization. Therefore, movements in dynamic sitting are caused by muscle activation. Whereas the CPM of the motorized chair is responsible for the movement. SEMG measurements were performed to determine whether muscle activity is detectable. The *m. rectus abdominus* (REC) was almost inactive during both conditions. Less activity of the abdominal muscles is assumed when sitting [73]. In contrast to REC a small activation of the lumbar *m. erector spinae* (LES) was found. However, only the number of activations was higher for MC. These findings are in line with the systematic review of O'Sullivan et al. [74]. Five of the seven studies included could not find any difference between dynamic and static sitting. Some changes in trunk muscle activity were found in two studies with exercise balls as dynamic sitting devices [75, 76]. They reported also increased discomfort, greater fatigue, and greater spinal shrinkage. McGill et al. [77] used an exercise ball but could not find differences in muscle activation. O'Sullivan et al. [74] concluded that dynamic sitting seems to be not sufficient to alter trunk muscle activity. Furthermore, greater muscle activity does not necessarily have to be associated with beneficial

outcomes. Most of the studies considered the mean muscle activity of the two conditions. This is problematic because activity changes in between are neglected. Our findings show the same mean muscle activity for both conditions. But the number of activations is higher for MC. This result is contradictory to O'Sullivan's conclusion. However, we assume that CPM has beneficial effects. The *m. erector spinae* consists of multiple small muscles that stabilize the spinal column. Segmental stability is considered a factor in back pain prevention [78]. It is reported that activation of LES in the range of 1–3 MVIC% is sufficient for stabilization of the spinal system [79]. The amplitude of a mean activation was ~ 3 MVIC% for both conditions. Therefore, the high amount of activations for MC may contribute to spinal stability. Whereby long-term effects have not yet been studied and are therefore uncertain.

### Spinal posture

The CPM did not affect the spinal posture regarding the lumbar lordosis angle, which did not differ for the two conditions. This is a neutral result. Some authors considered extended positions beneficial [24]. However, the individually preferred lumbar lordosis seems to be preventive [26]. Our findings show large deviations of the mean lumbar lordosis angle, which could be attributed to the different anatomically shaped lordosis (Fig 5). We found significantly less movement for SC (45 for 20 minutes, corresponding to 135 movements/h) compared to MC (498, respectively 1494 movements/h). Bontrup et al. [63] measured with a different device ~97 movements/h for SC. Pries et al. [60] investigated dynamic and static sitting using the Epionics SPINE system. They counted movements greater than 5°. This resulted in ~50 movements/h for SC and ~90 movements/h for sitting on an exercise ball. Even though the mean amplitude of movements in our study was lower than 5°, the number of movements is ten times higher for MC compared to SC. Whereas the exercise ball caused twice as many movements compared to SC. The high number of movements and the total lumbar lordosis change for MC shows that the CPM of the motorized chair increases spinal motion. This result was also stated for dynamic sitting [42].

### Physiological benefits of CPM

Variable loads and increased spinal motion are considered to be key factors in back health [28]. Mammalian experiments showed beneficial effects of spinal movement on intervertebral disc (IVD) nutrition [80, 81]. Further, spinal shrinkage is caused by fluid flow from the IVD. CPM can reduce the spinal shrinking, which can be explained by i) increased stiffness of the annulus fibrosus caused by changes in stress and relaxation or ii) a reduced fluid flow from the nucleus pulposus [33]. Comparable results were found for sitting on a dynamic chair [82] while a study with exercise balls showed contrary results [75]. Even though the causality could be more investigated [42], a dynamic sitting is recommended [25]. The participants showed different sitting behavior. Some were not affected by the chair, whereas others changed their habits due to the movement of the chair (Fig 5; ID01). The reason for this change is unclear. However, it appears that the CPM does not induce these participants to move. The movement starts with the onset of SC. This could be due to perceived discomfort for SC but should be further investigated. An example of static sitting can be derived from ID04 (Fig 5). This behavior is typical for people with LBP. They show more frequent static, end-range postures with less small and more large infrequent movements [83, 84]. The chair could be useful for people with static sitting behavior and/or LBP. In addition, the frequent changes in the lumbar lordosis caused by CPM could also be beneficial for those with variable posture. The motorized chair could be optimized with pressure sensors to detect the sitting behavior. A screening of the user would enable individualized CPM. Advantages could be the detection of long uninterrupted

periods, infrequent positions, and the magnitude of movements. Based on this information on- and offset timepoint, movement profile, and amount of distortion of the CPM could be assessed. Individualization of the CPM should be further investigated.

## Movement counts

The effect of the CPM on lumbar lordosis and the onset of the activated muscles is visualized in Fig 5. This allows inferring that high flexion angles occur when the seat is in the forward position. This is logical because subjects were then in a reclined position which is associated with high flexion angles [62]. In addition, the interaction between muscles and movements can be assessed. The highest muscle activity is observed when LES is in the most approximated position. The difference between the number of movements (498) and activations (~203) could be explained by only one peak activation during a cycle. The visualization shows moderate muscle activity and a smaller peak on the contralateral side, which was not detectable in some cases.

## Potentially adverse effects

Using a motorized chair could eventually lead individuals to change their sitting behavior, e. g. increasing the sitting duration or reducing the involuntary movements which could be seen in the movement pattern of ID 01 (Fig 5). It is generally important that people have an understanding of healthy sitting habits. However, these aspects should be given special consideration when using a chair with CPM.

Biomechanical stressors such as shear stress can be caused by the CPM. However, shear stress, lateral bending and axial rotation do not necessarily and directly lead to detrimental effects because they occur also in everyday life activity [85]. The high amount of short movements might cause tissue overloading, e.g. sustained contractions above 5% MVC can lead to muscle fatigue and pain [86].

## Limitations

Participants were young, healthy, and physically active. The results found are therefore only representative of the group described. LBP was neither an inclusion nor exclusion criterium. LBP was not predominant in this study, only two subjects had moderate to high pain intensity levels. Clear inclusion criteria could prevent the potential influence of LBP. A higher effect of the CPM was assumed, hence the priori power analysis led to the sample size of 20. However, this study enables power calculations for future studies. A limitation of the study was that the subjects were tested at different times of the day. Regarding the trunk kinematics, only the lumbar spine was assessed. No conclusions can be drawn about the thoracic and cervical spine. Also, lateral flexion and rotation of the spine were not investigated. Farina et al. [87] postulated some issues with sEMG measurement of the trunk muscles. They reported poor signal quality and less sensitivity of surface EMG on trunk muscles with complex architecture (e.g. LES). The main limitation of the sEMG measurement in our study was the inconsistent data for left and right LES. Even though the number of activations for CPM was higher, the mean activation remained comparable. The movement profile of the chair was chosen regarding the investigation of the child's gait [48]. Additionally, the movement was greater than in Lengsfeld's [40] study. Our result was observed in a particular setting. Future studies could therefore focus on different CPM profiles.

## Conclusion

The CPM of the motorized chair caused frequent low changes in lumbar lordosis angle and an increased number of muscle activations. Following the recommendations of variable and dynamic sitting [25], motorized chairs could be especially suitable for persons with static sitting behavior. The CPM did not affect average spinal posture, energy expenditure, or attentional control. This study gives first insights into CPM with larger deflection. However, potential positive or negative health effects were not evaluated. Future research is needed to assess the impact on health outcomes for musculoskeletal diseases like low back pain.

## Author Contributions

**Conceptualization:** Hendrik Schäfer, Petra Platen.

**Data curation:** Hendrik Schäfer, Robin Schäfer.

**Formal analysis:** Hendrik Schäfer, Robin Schäfer.

**Investigation:** Hendrik Schäfer.

**Methodology:** Hendrik Schäfer.

**Project administration:** Petra Platen.

**Software:** Hendrik Schäfer, Robin Schäfer.

**Visualization:** Hendrik Schäfer.

**Writing – original draft:** Hendrik Schäfer.

**Writing – review & editing:** Hendrik Schäfer, Robin Schäfer, Petra Platen.

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
