## [Decision Letter · Decision Letter 0]

23 Jan 2023

PONE-D-22-27895A novel motorized office chair causes low-amplitude spinal movements and activates trunk muscles: a cross-over trialPLOS ONE

Dear Dr. Schäfer,

Thank you for submitting your manuscript to PLOS ONE. After careful consideration, we feel that it has merit but does not fully meet PLOS ONE’s publication criteria as it currently stands. Therefore, we invite you to submit a revised version of the manuscript that addresses the points raised during the review process.

We look forward to receiving your revised manuscript.

Kind regards,

Friederike Schömig, M.D.

Academic Editor

PLOS ONE

Journal Requirements: 

"The chair in this study was a prototype developed and provided by the start-up company Vintus. The authors did not receive any funding to conduct the study. All authors declare, that the study is rigorously performed and transparently reported."

We note that you received funding from a commercial source: Vintus

Reviewers' comments:

Reviewer's Responses to Questions

**Comments to the Author**

1. Is the manuscript technically sound, and do the data support the conclusions?

Reviewer #1: Partly

Reviewer #2: Partly

2. Has the statistical analysis been performed appropriately and rigorously? 

Reviewer #1: Yes

Reviewer #2: I Don't Know

3. Have the authors made all data underlying the findings in their manuscript fully available?

Reviewer #1: Yes

Reviewer #2: Yes

4. Is the manuscript presented in an intelligible fashion and written in standard English?

Reviewer #1: Yes

Reviewer #2: Yes

5. Review Comments to the Author

Reviewer #1: This paper describes a single session, repeated measures, crossover study that analyzes the effects of a novel motorized office char on muscle activity as well as spinal posture and oxygen intake. In brief, the authors noted low changes in spinal posture and some muscle activation.

It is a well designed study, yet there some methodical flaws, that make a consideration for PLOS One questionable for this version.

First, the authors do not give detailed information about the participant’s demographics. This limits the quality of the study as it has been proven that the spinal column and especially lumbar lordosis are significantly effect by age and gender. Furthermore, by not further defining the study group it is questionable in how far the results are valid for everybody.

Second, the effect on muscle activation on the stabilization of spinal function is only assessed superficially. As the MVIC in both groups were 3%, the advantage of a more technologiccaly diversed product for everyday office life is questionable.

I recommend, in order to consider a publishing in PLOS One, that the author’s thoroughly revise their paper regarding the mentioned topics before re-submission.

Reviewer #2: The presented manuscript describes a randomized crossover study over the timeframe of 1h on a new kind of motorized office chair and its effects on sagittal lumbar movement and posture, paraspinal muscle activity, attentional control and oxygen uptake.

The manuscript is mostly well written and the topic of preventive interventions in long term-sitting is highly relevant on a societal level. The multi-parameter analysis including attentional control, oxygen uptake and trunk muscle activity and spinal posture is very interesting.

Nevertheless, there are major concerns regarding the following aspects:

1. The biomechanical reasoning for the described CPM-pattern of a lying eigth is not sufficiently described. Based on the understanding of the author of the described CPM pattern, it could potentially lead to negative effects on spinal degeneration (1-4) but a sufficient analysis is lacking:

a. A biomechanical analysis of assumed effects of this pattern on spinal movement – also in comparisation and loads e.g. through in sililco simulation is lacking

b. The lying eight most likely will induce shear stress, lateral bending and eventually axional rotation, a complex movement potentially associated with induction of enhanced degeneration

c. For induction of changes in lumbar flexion/extension without shear stress, a simple sagittal change of the angulation of the sitting surface would potentially be more efficient with fewer potentially negative effects

2. the scientific profoundness of the experimental set-up as it is lacking a relevant negative and positive control based on the 1h cross-over design. As the 1h design can be argued for the complex analysis for attentional control, oxygen uptake and EMG measurements, the measurements for lumbar posture are easily applicaple to a longer time frame and the profoundness of this data would critically improve with real-life negative control (e.g. e.g. standard office-chair with movement options) and real-life positive controls (e.g. working on a standing desk) which are already implemented in the prevetion of low back pain.

Minor concerns:

• The introduction would benefit from a brief description of behavioral aspects of sitting and the (not yet sufficient) understanding of different lumbar load and movement patterns and lumbar degeneration.

• The statistical analysis with simple t-test for longitudinal recorded data in multiple subjects in a crossover-design seems a little bit simplified, biostatistical counceling is recommended.

• Additional real-life data on CPM-Chair vs a real negative control and a positive control over an 8h workday with the epionic spine system and an actimeter would add additional value regarding the translationability of the observed effects.

• A critical discussion of potentially behavioral effects such as the “rebound effect” of a CPM chair on the working-day e.g. leading potentially to longer sitting phases etc is lacking .

• A critical discussion of the potential biomechanical effects of the CPM pattern on the lumbar spine is completely lacing.

• A critical discussion of potential different effects of different CPM patterns is lacking.

• The following sentence in the discussion does not work: “A priori power analysis based on small pilot studies”

• In the conclusion it should be also stated, that potential negative health effects were not evaluated.

Therefore I recommend major revisions with at least additional data and analysis on the potential effects of the CPM-Pattern on spinal biomechanics and loading, a critical reflection of these aspects and the above mentioned commentaries in the discussion and biostatistical counseling.

Furthermore additional data on spinal posture, breaks in sitting and movement (eg measured by actometers) over a working day with real life negative and positive controls and patient reported outcome measurements regarding low back pain, well-being at the workplace, subjective fatiguabilit would ad to the scientificall profoundness of the study.

6. PLOS authors have the option to publish the peer review history of their article (what does this mean?). If published, this will include your full peer review and any attached files.

Reviewer #1: No

Reviewer #2: No

---

## [Author Response · Author response to Decision Letter 0]

13 Mar 2023

Journal Requirements: 

Answer:

Thank you for the advice! We have carefully checked the manuscript and adapted it to the style requirements.

"The chair in this study was a prototype developed and provided by the start-up company Vintus. The authors did not receive any funding to conduct the study. All authors declare, that the study is rigorously performed and transparently reported."

We note that you received funding from a commercial source: Vintus

Answer:

We apologize for the inaccurate wording. We have not received funding from any commercial entity. Petra Platen has received a grant from the EU within the ERDF program. Two engineers have been employed to develop the prototype. We have elaborated on this circumstance extensively and included the desired statement. The amended Competing Interests Statement was added to the cover letter.

 

Review Comments to the Author:

Reviewer #1:

This paper describes a single session, repeated measures, crossover study that analyzes the effects of a novel motorized office char on muscle activity as well as spinal posture and oxygen intake. In brief, the authors noted low changes in spinal posture and some muscle activation. It is a well designed study, yet there some methodical flaws, that make a consideration for PLOS One questionable for this version.

First, the authors do not give detailed information about the participant’s demographics. This limits the quality of the study as it has been proven that the spinal column and especially lumbar lordosis are significantly effect by age and gender. Furthermore, by not further defining the study group it is questionable in how far the results are valid for everybody.

Answer:

Thank you for pointing out that the participant’s characteristics did not become clear for the reader. We agree that these details are highly important for interpreting to whom the study results might be representative for. Therefore, we included a participants characteristics table. A more detailed impression of the individual participants can be found in the comprehensive table in our repository: https://osf.io/7szrq.

In addition, we addressed the type of study population in the limitations:

”Participants were young, healthy, and physically active. The results found are therefore only representative of the group described”

Second, the effect on muscle activation on the stabilization of spinal function is only assessed superficially. As the MVIC in both groups were 3%, the advantage of a more technologiccaly diversed product for everyday office life is questionable. I recommend, in order to consider a publishing in PLOS One, that the author’s thoroughly revise their paper regarding the mentioned topics before re-submission.

Answer:

It is important to point out the limitations of superficial assessment of muscle activation. Thank you for raising this point! We have taken up this limitation and pointed out the problem of muscles with complex architecture such as the lumbar erector spinae:

Farina et al. [85] postulated some issues with sEMG measurement of the trunk muscles. They reported poor signal quality and less sensitivity of surface EMG on trunk muscles with complex architecture (e.g. LES). The main limitation of the sEMG measurement in our study was the inconsistent data for left and right LES.

It is true that both groups had similar mean activation of the lumbar muscles. However, the number of activations is higher for the moved condition. We address these different findings in detail in the discussion. We appreciate your concerns about the impact on the everyday office life. Therefore, we have extended the limitations:

Even though the number of activations for CPM was higher, the mean activation remained comparable.

Reviewer #2:

The presented manuscript describes a randomized crossover study over the timeframe of 1h on a new kind of motorized office chair and its effects on sagittal lumbar movement and posture, paraspinal muscle activity, attentional control and oxygen uptake. The manuscript is mostly well written and the topic of preventive interventions in long term-sitting is highly relevant on a societal level. The multi-parameter analysis including attentional control, oxygen uptake and trunk muscle activity and spinal posture is very interesting.

Nevertheless, there are major concerns regarding the following aspects:

1. The biomechanical reasoning for the described CPM-pattern of a lying eigth is not sufficiently described. Based on the understanding of the author of the described CPM pattern, it could potentially lead to negative effects on spinal degeneration (1-4) but a sufficient analysis is lacking:

a. A biomechanical analysis of assumed effects of this pattern on spinal movement – also in comparisation and loads e.g. through in sililco simulation is lacking

b. The lying eight most likely will induce shear stress, lateral bending and eventually axional rotation, a complex movement potentially associated with induction of enhanced degeneration

c. For induction of changes in lumbar flexion/extension without shear stress, a simple sagittal change of the angulation of the sitting surface would potentially be more efficient with fewer potentially negative effects

Answer:

Thank you for your kind words and your interest in our research. Your comments are provided on a well-developed understanding of biomechanics and (patho-)physiology of the spine and add important aspects to consider in this manuscript and in subsequent studies. 

Our response to comments a)-c) as follows:

a) We agree that a computer simulation would be helpful to determine potential loads created by different movement patterns. This study could have been done before our study. However, we consider our study as in important pilot study which informed our now ongoing research. We take your advice into account and are looking forward to conducting an in silico study to inform our research. However, we feel that this would be a separate study to conduct. Our current manuscript is rather long due to the many parameters investigated, so we would refrain from including this study in the current manuscript.

b) Like mentioned a lot along your review, we lack a thorough investigation of possible detrimental effects. Shear stress and other kinematic parameters could be assessed more thoroughly by a computer simulation (see a)) or mathematical modelling with experimental data. However, shear stress, lateral bending and axial rotation do not necessarily and directly lead to detrimental effects because they occur also in everyday life activity (Kripa & Kaur, 2021; Swain et al., 2020). Overuse of any such movement – even of low intensity –might trigger pain mechanisms such as peripheral sensitization leading to tissue overloading, e.g. sustained contractions above 5% MVC can lead to muscle fatigue (van Dieën et al., 2009) and pain (Björkstén & Jonsson, 1977) as stated by (Reeves et al., 2019). To address your concerns, we added these aspects to the “Potentially adverse effects” section. However, we agree that there is a need to quantify the relationship of spinal loads with data, which we aim to do in subsequent studies.

c) Thank you for sharing your idea! This protocol might be a suitable alternative when individuals with low back pain and/or when they experience problems with other protocols like the “8” we investigated in this study. We take your suggestion into account in our subsequent research!

2. the scientific profoundness of the experimental set-up as it is lacking a relevant negative and positive control based on the 1h cross-over design. As the 1h design can be argued for the complex analysis for attentional control, oxygen uptake and EMG measurements, the measurements for lumbar posture are easily applicaple to a longer time frame and the profoundness of this data would critically improve with real-life negative control (e.g. e.g. standard office-chair with movement options) and real-life positive controls (e.g. working on a standing desk) which are already implemented in the prevetion of low back pain.

Answer:

Thank you again for providing thoughts on possible research designs. Again, we value your ideas that we aim to involve in our ongoing, subsequent research. However, we hope that the reviewers end the editor agree with us, that these are subjects of new studies to conduct. The data presented in this manuscript provided some of these implications, e.g. it might be beneficial to investigate the biomechanical effects with a longer duration and add adequate controls. Thus, this study adds value to the scientific, accumulate knowledge and informs future designs.

Minor concerns:

• The introduction would benefit from a brief description of behavioral aspects of sitting and the (not yet sufficient) understanding of different lumbar load and movement patterns and lumbar degeneration.

Answer:

Thank for you suggestions to improve the introduction! We included brief introducing words at the beginning on sedentary behavior and the influence on health and wellbeing. Beside a short description of static and dynamic sitting behavior in the introduction, individual sitting behavior is discussed under the section “Physiological Benefits of CPM”.

We are convinced that the biomechanical aspects of sitting are important for understanding the effects of CPM. Therefore, we have included a section on the biomechanical effects of dynamic sitting in the introduction.

• The statistical analysis with simple t-test for longitudinal recorded data in multiple subjects in a crossover-design seems a little bit simplified, biostatistical counceling is recommended.

Answer:

Thank you for this advice. A colleague with background in mathematics (though, not a biostatistician) gave his feedback and approval. In our analyses, we followed the approach outlined by (Wellek & Blettner, 2012), which incorporates testing procedures for treatment and period (carry-over) effects. Since our study design is as simple as outlined in the study, we deem these tests as appropriate. Since we extracted features from the time-series data and time is treated as a discrete variable in the t-tests as condition (which is a common procedure), we also present the time-series data descriptively in figures 4 and 5 to convey important information of the qualitative impact of the moving chair.

• Additional real-life data on CPM-Chair vs a real negative control and a positive control over an 8h workday with the epionic spine system and an actimeter would add additional value regarding the translationability of the observed effects.

Answer:

Thank you for pointing out this research design as well. We agree that this study design is important and are looking forward to report in this issue in our subsequent study reports.

• A critical discussion of potentially behavioral effects such as the “rebound effect” of a CPM chair on the working-day e.g. leading potentially to longer sitting phases etc is lacking .

Answer:

We are pleased about raising the point that the behavior could be influenced by the CPM or the chair itself. We take a critical stance on this in the “Potentially Adverse Effects” section

• A critical discussion of the potential biomechanical effects of the CPM pattern on the lumbar spine is completely lacing.

Answer:

 We are thankful for your advice to improve the discussion. We have added a biomechanical explanation for the effects of the CPM. Potential adverse biomechanical effects are covered by the corresponding section.

• A critical discussion of potential different effects of different CPM patterns is lacking.

Answer:

Thank you for pointing out the importance of an investigation and discussion of different CPM patterns. We refer to the individualization of the CPM in the discussion and focusing on one CPM pattern is mentioned as a limitation. As already mentioned, we gratefully accept your suggestion and will take it into account in future studies.

• The following sentence in the discussion does not work: “A priori power analysis based on small pilot studies”

Answer:

 The sentence is excluded, and the information is added to the following sentence, thank you!

• In the conclusion it should be also stated, that potential negative health effects were not evaluated.

Answer:

 We added your suggestion, thanks!

Therefore I recommend major revisions with at least additional data and analysis on the potential effects of the CPM-Pattern on spinal biomechanics and loading, a critical reflection of these aspects and the above mentioned commentaries in the discussion and biostatistical counseling. Furthermore additional data on spinal posture, breaks in sitting and movement (eg measured by actometers) over a working day with real life negative and positive controls and patient reported outcome measurements regarding low back pain, well-being at the workplace, subjective fatiguabilit would ad to the scientificall profoundness of the study.

Answer:

Thank you once more for your intellectual contribution! Indeed, we now receive funding for three additional years to investigate long-term effects of various CPM and also stochastic movement patterns where we plan to do field studies, investigate LBP populations and relationships with additional factors such as stress. Your comments are very helpful and partly align with our plans! Again, we hope to convince you that this study provides preliminary data which is important to inform future research and, thus, adds to the knowledge of the scientific community.

---

## [Decision Letter · Decision Letter 1]

7 Aug 2023

PONE-D-22-27895R1A novel motorized office chair causes low-amplitude spinal movements and activates trunk muscles: a cross-over trialPLOS ONE

Dear Dr.  Schäfer,

Thank you for submitting your manuscript to PLOS ONE. After careful consideration, we feel that it has merit but does not fully meet PLOS ONE’s publication criteria as it currently stands. Therefore, we invite you to submit a revised version of the manuscript that addresses the points raised during the review process.

We look forward to receiving your revised manuscript.

Kind regards,

Monika Błaszczyszyn

Academic Editor

PLOS ONE

Reviewers' comments:

Reviewer's Responses to Questions

**Comments to the Author**

1. If the authors have adequately addressed your comments raised in a previous round of review and you feel that this manuscript is now acceptable for publication, you may indicate that here to bypass the “Comments to the Author” section, enter your conflict of interest statement in the “Confidential to Editor” section, and submit your "Accept" recommendation.

Reviewer #3: (No Response)

Reviewer #4: All comments have been addressed

2. Is the manuscript technically sound, and do the data support the conclusions?

Reviewer #3: Partly

Reviewer #4: Partly

3. Has the statistical analysis been performed appropriately and rigorously? 

Reviewer #3: No

Reviewer #4: Yes

4. Have the authors made all data underlying the findings in their manuscript fully available?

Reviewer #3: Yes

Reviewer #4: Yes

5. Is the manuscript presented in an intelligible fashion and written in standard English?

Reviewer #3: Yes

Reviewer #4: Yes

6. Review Comments to the Author

Reviewer #3: Thank you for giving me the opportunity to review this interesting paper on a novel motorized chair that gave an interesting insight into movement, energy expenditure and attention during static and dynamic sitting. While the paper makes for interesting reading, I have some concerns, mostly related to the methodology, which need to be addressed before the manuscript could be considered for publication.

Introduction:

In the third paragraph, the authors correctly point out that there is no direct link between back pain and sitting time, possibly due to the multifaceted nature of backpain. That being said, it comes a bit as a surprise that, in the following sections, the authors seem to link most of their reasoning for the conduction of this study to back pain. Please consider rewriting as evidence suggests that there are other fields of health promotion, such as sedentary behavior, that could serve as a stronger base for the justification of this study.

In the second to last paragraph, the authors use the phrase “positive effects of a passive moved sitting”. Please spell these out and relate them to your outcome measures as it makes for somewhat cumbersome reading if these positive effects have to searched for in different parts of the manuscript.

The outcome attention is not introduced in the introduction, therefore a justification of investigating this outcome is missing, weakening the rationale of the study.

Methods:

Study design:

First paragraph and in general: The outcomes should always be listed in the same order throughout the manuscript as this improves readability.

The inclusion and exclusion criteria seem very brought. For example: what does any health restriction precisely refer to? For example, if a participant had a cough, was the participant excluded? Please list them precisely to make your study more reproducible. Furthermore, some outcome measures are sensitive to anthropometrics: For example sEMG and measurements of spinal kinematics are influenced by the BMI. Was there an upper and lower limit for the BMI?

The outcome attentional control comes a bit out of the blue, kindly introduce and justify it in the introduction.

“The hip and knee angles were set to 90°.” How was this operationalized?

Participants:

To which guidelines are you referring to with reference 46? This sentence is not self-explanatory.

Motorized chair:

What was the rationale to set the movement speed to 56 mm/s.?

Oxygen uptake:

were different sizes of oxygen masks available?

Trunk muscle activity:

The authors state the Myon system is reliable. to what type of reliability does this refer? Inter-rater (placement of electrodes) intrarater? this is not self-explanatory.

Data processing:

“The data were smoothed with a slightly moving average” what does slightly refer to, does this refer to the sEMG moving average, the epionics or something else? If the epionics and sEMG were put in relation to each other than how were they synchronized timewise, regarding aspects such as the different measurement frequencies, possible delays or inconsistent time stamps?

“Amplitudes greater than 1° were considered for further analysis”. is 1° greater than the measurement error of the device?

“repetitive movement of the chair leads to similar positive and negative angle changes” Does this imply that they are always similar in magnitude?

“To estimate the individual total movement in each condition, the number of movements

was multiplied by the average movement amplitude“ why was it multiplied by the average, why not make a sum of all peak to peak amplitudes? kindly explain

“The sEMG and Epionics data were synchronized to assess the spinal movements and their corresponding muscle activity.” how often were they synchronized? How many recordings were done per participant and how did you handle the manual synchronization given the two different recording frequencies? did you upsample or downsample one of the time series?

“By tapping the sensors twice, peaks are generated that allow data synchronization.” First, I would describe this in the beginning of this section as it is important to understand how you performed your measurements. second please explain why this method is valid considering that you have different recording frequencies, also in light of other potential sources of bias such as delays in recordings.

Statistical analysis:

you did test for the normality of the data distribution so what was your rationale for the t-tests? also explain your reasoning for using an unpaired t-test in light of the argument that Paired t-test is used to compare the means of two samples when each individual in one sample also appears in the other sample.

“Non-normally distributed data were analyzed unchanged.” please explain why you checked the distribution if it had no consequence? what is your rationale for not switching to non-parametric tests?

Results:

consider using the same order of outcomes in each section, as this greatly helps the reader.

Trunk muscle activity

“Data inconsistencies occurred quite regularly (e.g., large differences between left and right side, low signal), but we did not exclude these data.” why not, what was your rationale for not considering these as measurement errors or "bad data"? how do you justify this given the assumption that these might influence your results and perhaps lead you to wrong conclusions?

Spinal posture:

Figure 5b shows that the static movement is rather random, compared to the cyclic movement in 5a. can you justify that the methods used to derive peak to peak amplitude in a cyclic movement are also valid for the more random movement observed in 5b? How do you explain that the three subjects in figure 5 show a very different basic lordosis angle, can this be due to placement or measurement error?

how does the small range of motion compare to the measurement error or placement error?

Discussion

Attentional control:

“After the removal of an outlier, the average performance for MC was 1 % higher(p = .059).» is this a relevant difference, also regarding intra-subject variance and all other factors that can influence attentional control that you might not have been able to control?

Muscle activity

“Therefore, movements in dynamic sitting are caused by muscle activation.” sentence seems misleading, if meaning is that only motorized seats can cause additional muscle activation, kindly refer to https://pubmed.ncbi.nlm.nih.gov/35905083/ (pelvis initiated exercises)

Otherwise kindly explain if you imply that the muscular activity could not also be reactive to the moving seat.

¨The m. rectus abdominus (REC) was almost inactive during both conditions.» was it inactive or was the activity not detectable?

“Segmental stability is considered an important factor in back pain prevention.” given the heterogeneity of LBP, is this statement a bit too brave? can you elaborate on which type of LBP sufferers might benefit from active sitting?

Posture

“The CPM did not affect the spinal posture regarding the lumbar lordosis angle, which did not differ for the two conditions.” With the information provided in Figure 5: the mean angle did not differ much, but the angular displacement is very different (cyclic vs random movement).

“Our findings show large deviations of the mean lumbar lordosis angle, which could be attributed to the different anatomically shaped lordosis (Fig. 5).” this could be verified by investigating neutral sitting or standing, are there data on neutral sitting perhaps from epionics or from video? if yes, kindly verify this assumption as it would also help to ensure that this not due to a measurement or placement error.

Physiological benefits of CPM

“They show more frequent static, end-range postures with less small and more large

infrequent movements « Figure 5: ID04 does not look like an end of range posture, how many participants did assume an end range posture?

“A screening of the user would enable individualized CPM.” while this is an interesting thought, this is nevertheless a frequent and often unproven claim made for any novel health technology. how should an individualized CPM be derived from screening of which parameters?

Movement counts:

“In addition, the interaction between muscles and movements can be assessed. The highest muscle activity is observed when LES is in the most approximated position. The difference between the number of movements (498) and activations (~203) could be explained by only one peak activation during a cycle.” A graph illustrating this would be very interesting, however given the methodological reservation expressed earlier it first needs to be explained by the authors how they synchronized the time series in more detail.

Reviewer #4: Dear Authors,

Greetings.

Thank you for the opportunity to review the paper titled “A novel motorized office chair causes low-amplitude spinal movements and activates trunk muscles: a cross-over trial”.

Article ID. PONE D- 22- 27895-R1”

I appreciate the authors’ efforts in conducting this work on Twenty office workers.

The authors studied a single session (30 min) of continuous passive motion (CPM) by a novel motorized office chair on multiple variables and found slight changes in lumbar lordosis angle.

Overall, the paper is well written, but I have a few concerns about the design, sample selection and size and methodology.

Please find my review comments below.

In introduction,

In the initial paragraphs, the authors introduced sedentary sitting behaviour and physical inactivity and its impact on health-related issues such as low back pain (LBP), diabetes and cardiovascular risks. In the round-up, they stressed the importance of dynamic sitting and loading conditions of the vertebral column. Later, they introduce the advent of motorized chairs with continuous passive motion (CPM) and its effect on LBP. The authors rationalize that less passive movement (axial rotation of 1.6° from Lengsfeld's study) is insufficient to produce a positive effect, warranting a motorized chair with a greater deflection of translatory movement. Therefore, to investigate the benefits of CPM in sitting on lumbar lordosis and trunk muscle activation (rectus abdominus and erector spinae), as well as oxygen uptake in healthy adults.

The concept of dynamic sitting and its spinal loading has already been documented elsewhere, as well as strategies to prevent and manage LBP. Please motivate me why/how a CPM-motorized chair could be another potential option for trunk muscle activity (reactive) compared to person-preferred dynamic sitting (anticipatory).

If the authors address the need to conduct this study citing the limited axial rotation of 1.6° (Lengsfeld), I believe this novel motorized chair should preferably be tested in subjects with LBP. I’m just wondering if the participants are young adults (mean age of 24.5 ± 3.6 28.6 ± 3.3 years) with moderate physical activity (IPAQ moderate 1320 ± 1478; 975 ± 749), the subject selection in this study is not well aligned with the need of the study.

Methods,

A crossover design is usually chosen when the required sample size may not be met in clinical studies. Otherwise, I’m fine with the current study design. Comparing two healthy groups (one sitting on a motor chair, another group on a stable chair) may be an option!!

Also, it’s quite difficult to interpret the findings just with one session. It’s known that immediate muscle response and muscle activity do happen when someone is sitting on a moving platform. What is more interesting is how muscle activity is when someone is seated on MC for a long time. This may have a clinical implication, particularly for preventing musculoskeletal dysfunction perspective (LBP, posture etc.).

Measuring the attention control is more like a feasibility aspect of a MC whether it disturbs the attentional ability of participants. Would it be ideal to see through long working hours!! Individuals tend to adapt to postural and gaze stability when seated/standing on a low-amplitude moving platform!! Please motivate why was oxygen uptake a measure!! Would an hour sitting on a chair demand more MET!!

The authors studied only two trunk muscle activity profile (lumbar erector spinae and m. rectus abdominus). They are the phasic muscles and postural muscles (multifidus, oblique) are the first ones to contract when there is a postural perturbation!! Therefore, it's difficult to generalize the findings!! I understand that the EMG unit should be more sensitive to pick up these myosignals. I appreciate the authors' efforts to study the trunk muscles' EMG profile.

The results, findings and discussion are well-written.

Often study weaknesses are missed opportunities, but not the limitations!!

Tables and figures are useful.

There are too many citations and reference papers can be cut short.

Overall, I liked the write-up and I appreciate authors’ efforts in conducting this work.

Best wishes.

Kind regards,

The reviewer.

7. PLOS authors have the option to publish the peer review history of their article (what does this mean?). If published, this will include your full peer review and any attached files.

Reviewer #3: No

Reviewer #4: **Yes: **Suruliraj Karthikbabu

---

## [Author Response · Author response to Decision Letter 1]

5 Oct 2023

Dear Dr. Monika Błaszczyszyn,

We would like to thank you for the feedback on our manuscript titled “A novel motorized office chair causes low-amplitude spinal movements and activates trunk muscles: a cross-over trial” and the opportunity to re-submit and address the comments made by the two expert reviewers. Also, we would like to thank the expert reviewers for taking their time to read and comment on our manuscript; the comments were clearly the result of a thorough and very insightful review. We feel that the revisions have added to the strength of the manuscript.

The specific reviewer comments have been addressed below. An amended version of the manuscript has been re-submitted, where changes have been highlighted in yellow. 

Kind Regards

Hendrik Schäfer in behalf of the authors

 

Review Comments to the Author:

Reviewer #3: Thank you for giving me the opportunity to review this interesting paper on a novel motorized chair that gave an interesting insight into movement, energy expenditure and attention during static and dynamic sitting. While the paper makes for interesting reading, I have some concerns, mostly related to the methodology, which need to be addressed before the manuscript could be considered for publication.

Answer:

Dear Reviewer #3, thank you for reviewing our manuscript and for providing valuble feedback!

Introduction:

In the third paragraph, the authors correctly point out that there is no direct link between back pain and sitting time, possibly due to the multifaceted nature of backpain. That being said, it comes a bit as a surprise that, in the following sections, the authors seem to link most of their reasoning for the conduction of this study to back pain. Please consider rewriting as evidence suggests that there are other fields of health promotion, such as sedentary behavior, that could serve as a stronger base for the justification of this study.

Answer:

Thank you for raising this point. We tried to present the current status of back pain and sitting in a neutral and objective way. This should be understood as contextualization and not as a justification for conducting the study.The reason for investigating CPM lies rather in the recommendation of variable sitting.

In the second to last paragraph, the authors use the phrase “positive effects of a passive moved sitting”. Please spell these out and relate them to your outcome measures as it makes for somewhat cumbersome reading if these positive effects have to searched for in different parts of the manuscript.

Answer:

Adding this information at the section in the introduction improves the readabilty of the manuscript.

The outcome attention is not introduced in the introduction, therefore a justification of investigating this outcome is missing, weakening the rationale of the study.

Answer:

We used the AX-CPT test for two reasons: First of all the test guarantees a highly standardized measurement situation. Second, in addition it gives us informations on the attentional control. We added the outcome to the introduction. 

Methods:

Study design:

First paragraph and in general: The outcomes should always be listed in the same order throughout the manuscript as this improves readability.

Answer:

Thank you for the advise. We change the section accordingly.

The inclusion and exclusion criteria seem very brought. For example: what does any health restriction precisely refer to? For example, if a participant had a cough, was the participant excluded? Please list them precisely to make your study more reproducible. Furthermore, some outcome measures are sensitive to anthropometrics: For example sEMG and measurements of spinal kinematics are influenced by the BMI. Was there an upper and lower limit for the BMI?

Answer:

Particpants who were able and allowed to work were included. This means that a patient with a cough which would not allow them to work would have been excluded.

We added this information to the manuscript.

There was no upper or lower limit for the BMI. 

The outcome attentional control comes a bit out of the blue, kindly introduce and justify it in the introduction.

Answer:

Thank you for the advise. We added the informations to the introduction.

“The hip and knee angles were set to 90°.” How was this operationalized?

Answer:

We measured the angle with a protector. The height of the chair was adjusted accordingly and smaller subjects used a footrest.

Participants:

To which guidelines are you referring to with reference 46? This sentence is not self-explanatory.

Answer:

Thank your for this advise! We added the information that we refer to the IPAQ guidelines.

Reference 46 is: IPAQ Research Committee. Guidelines for data processing and analysis of the International Physical Activity Questionnaire (IPAQ)-short and long forms. 2005 [cited 7 Sep 2020]. Available from: www.ipaq.ki.se.

Motorized chair:

What was the rationale to set the movement speed to 56 mm/s.?

Answer:

We choosed the maximal possible velocity of the protoype in order to produce as much as possible movements for the experiment. We added this information. 

Oxygen uptake:

were different sizes of oxygen masks available?

Answer:

Yes it was. We used Vyaire Vyntus CPX mask in size S and M to guarantee that the “the face mask with a filter was attached airtight”.

Trunk muscle activity:

The authors state the Myon system is reliable. to what type of reliability does this refer? Inter-rater (placement of electrodes) intrarater? this is not self-explanatory.

Answer:

It refers to intra-session and inter-day testing. We added the information. The information comes from the work of Sorbie et al.:

 Sorbie GG, Williams MJ, Boyle DW, Gray A, Brouner J, Gibson N, et al. Intra-session and Inter-day Reliability of the Myon 320 Electromyography System During Sub-maximal Contractions. Front Physiol. 2018; 9:309. doi: 10.3389/fphys.2018.00309 PMID: 29651252.

Data processing:

“The data were smoothed with a slightly moving average” what does slightly refer to, does this refer to the sEMG moving average, the epionics or something else? If the epionics and sEMG were put in relation to each other than how were they synchronized timewise, regarding aspects such as the different measurement frequencies, possible delays or inconsistent time stamps?

Answer:

The moving averages were used for both the sEMG and the Epionics dataset. 

We generated timestamps by tapping a the sensors twice:

Therefore, one sEMG sensor was attached to one Epionics accelerometer. By tapping the sensors twice, peaks are generated that allow data synchronization.

Then we used the table2timetable, synchronize and retime function in Matlab to align the data.

All our codes can be found in our repository (https://osf.io/7szrq/), which offers maximal transperency. 

“Amplitudes greater than 1° were considered for further analysis”. is 1° greater than the measurement error of the device?

Answer:

In the work of Consmüller et al. is shown that the difference for the flexion between Epionics and radiography is 1°.

“repetitive movement of the chair leads to similar positive and negative angle changes” Does this imply that they are always similar in magnitude?

Answer:

One movement was defined with this approach, therefore the destribution of the positive and negative magnitude is not that relevant. In addtion, we adapted the approaches of Zemp et al. and Bontrup et al.

“To estimate the individual total movement in each condition, the number of movements

was multiplied by the average movement amplitude“ why was it multiplied by the average, why not make a sum of all peak to peak amplitudes? kindly explain

Answer:

Thank you for this idea. This would be also a good and probaply a more precise way to report the total movement. However, our intention was to combine the given information, which are in the variables “number of movements” and “mean amplitude [°]”, to make the difference in the conditions more descriptive. We believe that outcome would not be influenced by the two calculation methods.

“The sEMG and Epionics data were synchronized to assess the spinal movements and their corresponding muscle activity.” how often were they synchronized? How many recordings were done per participant and how did you handle the manual synchronization given the two different recording frequencies? did you upsample or downsample one of the time series?

“By tapping the sensors twice, peaks are generated that allow data synchronization.” First, I would describe this in the beginning of this section as it is important to understand how you performed your measurements. second please explain why this method is valid considering that you have different recording frequencies, also in light of other potential sources of bias such as delays in recordings.

Answer:

The measurements were synchronized once in the beginning.

There was only one recording for each participant. 

We used the table2timetable, synchronize and retime functions in Matlab. The scripts can be found in our repository.

We believe that is essential to describe the processing of the data before the synchronization is mentioned. 

We checked if the method is valid by evaluating the timepoints of the on- and offset of the conditions, in order to account for delays in recordings.

Of note, the analysis of our data and the causal connection confirms that this method is valid (Figure 4).

Statistical analysis:

you did test for the normality of the data distribution so what was your rationale for the t-tests? also explain your reasoning for using an unpaired t-test in light of the argument that Paired t-test is used to compare the means of two samples when each individual in one sample also appears in the other sample.

“Non-normally distributed data were analyzed unchanged.” please explain why you checked the distribution if it had no consequence? what is your rationale for not switching to non-parametric tests?

Answer:

We strictly followed the instructions of Wellek and Blettner (2012):

Wellek S, Blettner M. On the proper use of the crossover design in clinical trials: part 18 of a series on evaluation of scientific publications. Deutsches Ärzteblatt International. 2012; 109:276–81. doi: 10.3238/arztebl.2012.0276 PMID: 22567063.

The explanation for using unpaired t tests can be found in the work of Wellek and Blattner: 

“The differences between treatment effects can be assessed by means of a standard t-test for independent samples using the intra-individual differences between the outcomes in both periods as the raw data.”

We calculated and reported the distribution in order to give this information to reader. 

We did not switched to nonparametric tests, to stay inline with the recommendations of Wellek and Blattner. 

The data and the calculations can be found in our repository. 

Results:

consider using the same order of outcomes in each section, as this greatly helps the reader.

Answer:

The order is:

• Methods

Attentional control, Oxygen uptake, Trunk muscle activity, Spinal posture

• Results

Attentional control, Oxygen uptake, Trunk muscle activity, Spinal posture, 

• Discussion

Attentional control, Energy expenditure, Muscle activity, Posture

Trunk muscle activity

“Data inconsistencies occurred quite regularly (e.g., large differences between left and right side, low signal), but we did not exclude these data.” why not, what was your rationale for not considering these as measurement errors or "bad data"? how do you justify this given the assumption that these might influence your results and perhaps lead you to wrong conclusions?

Answer:

We discussed this with our research team and concluded that we cannot assess whether this is physiological or due to measurement error. Therefore, we decided not to exclude the data. We are confident that this does not affect our conclusions as the discrepancies existed in both conditions.

Spinal posture:

Figure 5b shows that the static movement is rather random, compared to the cyclic movement in 5a. can you justify that the methods used to derive peak to peak amplitude in a cyclic movement are also valid for the more random movement observed in 5b? How do you explain that the three subjects in figure 5 show a very different basic lordosis angle, can this be due to placement or measurement error?

how does the small range of motion compare to the measurement error or placement error?

Answer:

It is justified by the way of processing the data. When the difference between two amplitudes was greater than 1° it was count as one movement. In addition, we conducted visual checks if all movements were detected. The codes for the figures can also be found in our scripts. 

“The mean lumbar lordosis angle was highly variable for MC 0.3° ± 14.9° and for SC 2.8° ± 15.4° (p = .058; d = .165).”

We argue: 

“Our findings show large deviations of the mean lumbar lordosis angle, which could be attributed to the different anatomically shaped lordosis (Fig. 5).”

Therefore, we do not consider placement or measurement errors as reasons for this difference. 

Discussion

Attentional control:

“After the removal of an outlier, the average performance for MC was 1 % higher(p = .059).» is this a relevant difference, also regarding intra-subject variance and all other factors that can influence attentional control that you might not have been able to control?

Answer:

No, however, we argue that the hypothesis from Lengsfeld et al. can not be confirmed with our observered results. Therefore we just state that there is no difference between the two conditions and hence no disadvantage of the moved conditions. 

Muscle activity

“Therefore, movements in dynamic sitting are caused by muscle activation.” sentence seems misleading, if meaning is that only motorized seats can cause additional muscle activation, kindly refer to https://pubmed.ncbi.nlm.nih.gov/35905083/ (pelvis initiated exercises)

Otherwise kindly explain if you imply that the muscular activity could not also be reactive to the moving seat.

Answer:

We just wanted to state that movements in dynamic sitting are caused by the use of muscle, whereas the CPM of the motorized chair is responsible for the movement. Since the CPM causes the movement the muscle activity is likely reactive to the moving seat.

¨The m. rectus abdominus (REC) was almost inactive during both conditions.» was it inactive or was the activity not detectable?

“Segmental stability is considered an important factor in back pain prevention.” given the heterogeneity of LBP, is this statement a bit too brave? can you elaborate on which type of LBP sufferers might benefit from active sitting?

Answer:

We conclude that the muscle was almost inactive because: “Less activity of the abdominal muscles is assumed when sitting. (O'Sullivan et al. 2006)”.

We deleted the “important” in order to account for the heterogeneity of LBP.

We discuss this in the benefit section: 

“(…)This behavior is typical for people with LBP. They show more frequent static, end-range postures with less small and more large infrequent movements [83,84]. The chair could be useful for people with static sitting behavior and/or LBP.”

Posture

“The CPM did not affect the spinal posture regarding the lumbar lordosis angle, which did not differ for the two conditions.” With the information provided in Figure 5: the mean angle did not differ much, but the angular displacement is very different (cyclic vs random movement).

“Our findings show large deviations of the mean lumbar lordosis angle, which could be attributed to the different anatomically shaped lordosis (Fig. 5).” this could be verified by investigating neutral sitting or standing, are there data on neutral sitting perhaps from epionics or from video? if yes, kindly verify this assumption as it would also help to ensure that this not due to a measurement or placement error.

Answer:

We did not assessed this. However we cite 5 articles from the research groups of Pries and Taylor were this data are available. For example the work of Consmüller (doi: 10.1007/s00586-012-2312-1) shows also different anatomically shaped lordosis and confirms the observation with other measurements (radiography).

Physiological benefits of CPM

“They show more frequent static, end-range postures with less small and more large

infrequent movements « Figure 5: ID04 does not look like an end of range posture, how many participants did assume an end range posture?

Answer:

It is right that participants with ID4 does not show an end range posture. We were rather referring to more frequent static, less small and more large infrequent movements.

We can not give any informations on end range posture since this was not assessed.

“A screening of the user would enable individualized CPM.” while this is an interesting thought, this is nevertheless a frequent and often unproven claim made for any novel health technology. how should an individualized CPM be derived from screening of which parameters?

 

Answer:

These are interesting questions that could be investigated in further studies. Our work provides a starting point for these considerations. In particular, the various patterns in Figure 5 are helpful for examining this.

Movement counts:

“In addition, the interaction between muscles and movements can be assessed. The highest muscle activity is observed when LES is in the most approximated position. The difference between the number of movements (498) and activations (~203) could be explained by only one peak activation during a cycle.” A graph illustrating this would be very interesting, however given the methodological reservation expressed earlier it first needs to be explained by the authors how they synchronized the time series in more detail.

Answer:

The information of the difference between the number of movements and activations is in the figure 4. In 4.a could only one clear peak be detected whereas in 4.b two peaks are shown. 

The synchronization was done with the matlab table2timetable, synchronize and retime function with an linear interpolation of the epionics data, due to the lower sampling rate.

Reviewer #4: Dear Authors,

Greetings.

Thank you for the opportunity to review the paper titled “A novel motorized office chair causes low-amplitude spinal movements and activates trunk muscles: a cross-over trial”.

Article ID. PONE D- 22- 27895-R1”

I appreciate the authors’ efforts in conducting this work on Twenty office workers.

The authors studied a single session (30 min) of continuous passive motion (CPM) by a novel motorized office chair on multiple variables and found slight changes in lumbar lordosis angle.

Answer:

Dear Prof. Dr. Karthikbabu, thank you for your kind words and your interest in our research. We appreciate your valuable feedback and we try to answer all of your questions. 

Overall, the paper is well written, but I have a few concerns about the design, sample selection and size and methodology.

Please find my review comments below.

In introduction,

In the initial paragraphs, the authors introduced sedentary sitting behaviour and physical inactivity and its impact on health-related issues such as low back pain (LBP), diabetes and cardiovascular risks. In the round-up, they stressed the importance of dynamic sitting and loading conditions of the vertebral column. Later, they introduce the advent of motorized chairs with continuous passive motion (CPM) and its effect on LBP. The authors rationalize that less passive movement (axial rotation of 1.6° from Lengsfeld's study) is insufficient to produce a positive effect, warranting a motorized chair with a greater deflection of translatory movement. Therefore, to investigate the benefits of CPM in sitting on lumbar lordosis and trunk muscle activation (rectus abdominus and erector spinae), as well as oxygen uptake in healthy adults.

The concept of dynamic sitting and its spinal loading has already been documented elsewhere, as well as strategies to prevent and manage LBP. Please motivate me why/how a CPM-motorized chair could be another potential option for trunk muscle activity (reactive) compared to person-preferred dynamic sitting (anticipatory).

Answer:

Thank you for raising this point! We are addressing this point in the discussion (Physiological benefits of CPM). In general, whenever there is an voluntary muscle contraction then this is superior to the reactive activation of muscle caused by the CPM. Therefore, it could be assumed that individuals with dynamic sitting behavior do not have the need for CPM then others with static sitting behavior.To conclude, CPM can be beneficial for individuals with static sitting behavior and low personal willingness to change their behavior.

If the authors address the need to conduct this study citing the limited axial rotation of 1.6° (Lengsfeld), I believe this novel motorized chair should preferably be tested in subjects with LBP. I’m just wondering if the participants are young adults (mean age of 24.5 ± 3.6 28.6 ± 3.3 years) with moderate physical activity (IPAQ moderate 1320 ± 1478; 975 ± 749), the subject selection in this study is not well aligned with the need of the study.

Answer:

We are thankful for this consideration, since we are planning to investigate this in our next studies. 

However, this is the first study of this novel chair and we decided to first evaluate the effects of the CPM on healthy individuals. Another minor reason was that due to Covid-19 restrictions, only employees in our department could be tested.

Methods,

A crossover design is usually chosen when the required sample size may not be met in clinical studies. Otherwise, I’m fine with the current study design. Comparing two healthy groups (one sitting on a motor chair, another group on a stable chair) may be an option!!

Answer:

Yes, this is totally right. However, we decided to use the single session design, in order to have the less potantially influencing factors in both conditions.

Also, it’s quite difficult to interpret the findings just with one session. It’s known that immediate muscle response and muscle activity do happen when someone is sitting on a moving platform. What is more interesting is how muscle activity is when someone is seated on MC for a long time. This may have a clinical implication, particularly for preventing musculoskeletal dysfunction perspective (LBP, posture etc.).

 

Answer:

This is definitley an interesting aspect and in scope of our further investigations. However, we state in the section “Muscle Activity” that “(..) long-term effects have not yet been studied and are therefore uncertain.”

Measuring the attention control is more like a feasibility aspect of a MC whether it disturbs the attentional ability of participants. Would it be ideal to see through long working hours!! 

Answer:

We cannot say that MC is in favor regarding the attentional control, but we were also surprised by the results. It would be definitely interesting to investigate longer duration, especially when the results of the work of Henz et al is taken into account: 

“They reported an improvement in mathematical performance during dynamic sitting”

“Henz et al. [68] observed longer durations of up to 4 hours (…)”

Individuals tend to adapt to postural and gaze stability when seated/standing on a low-amplitude moving platform!! Please motivate why was oxygen uptake a measure!! Would an hour sitting on a chair demand more MET!!

Answer:

We had the question if CPM would increase the demand over the treshold (MET ≥ 1.5) for sedentary behaviour, which was not confirmed. We argue that longer duration of CPM would not increase the oxygen uptake, since we could not detect an increase with the observerd 20 minutes. 

The authors studied only two trunk muscle activity profile (lumbar erector spinae and m. rectus abdominus). They are the phasic muscles and postural muscles (multifidus, oblique) are the first ones to contract when there is a postural perturbation!! Therefore, it's difficult to generalize the findings!! I understand that the EMG unit should be more sensitive to pick up these myosignals. I appreciate the authors' efforts to study the trunk muscles' EMG profile.

Answer:

It is right that it is difficult to assess the muscle activity with the sEMG. Therefore we tried to interpret the results with coution and added a critical discussion citing relevant articles such as the one of Farina et al. Thank you very much for honoring our work!

The results, findings and discussion are well-written.

Often study weaknesses are missed opportunities, but not the limitations!!

Tables and figures are useful.

There are too many citations and reference papers can be cut short.

Overall, I liked the write-up and I appreciate authors’ efforts in conducting this work.

Best wishes.

Kind regards,

The reviewer.

---

## [Decision Letter · Decision Letter 2]

9 Nov 2023

A novel motorized office chair causes low-amplitude spinal movements and activates trunk muscles: a cross-over trial

PONE-D-22-27895R2

Dear Dr. Schafer,

We’re pleased to inform you that your manuscript has been judged scientifically suitable for publication and will be formally accepted for publication once it meets all outstanding technical requirements.

Kind regards,

Monika Błaszczyszyn

Academic Editor

PLOS ONE

Additional Editor Comments (optional):

Reviewers' comments:

Reviewer's Responses to Questions

**Comments to the Author**

1. If the authors have adequately addressed your comments raised in a previous round of review and you feel that this manuscript is now acceptable for publication, you may indicate that here to bypass the “Comments to the Author” section, enter your conflict of interest statement in the “Confidential to Editor” section, and submit your "Accept" recommendation.

Reviewer #4: All comments have been addressed

2. Is the manuscript technically sound, and do the data support the conclusions?

Reviewer #4: Yes

3. Has the statistical analysis been performed appropriately and rigorously? 

Reviewer #4: Yes

4. Have the authors made all data underlying the findings in their manuscript fully available?

Reviewer #4: Yes

5. Is the manuscript presented in an intelligible fashion and written in standard English?

Reviewer #4: Yes

6. Review Comments to the Author

Reviewer #4: Dear authors,

I appreciate authors for revising the manuscript and making the necessary corrections in the text.

Best wishes

7. PLOS authors have the option to publish the peer review history of their article (what does this mean?). If published, this will include your full peer review and any attached files.

Reviewer #4: **Yes: **Suruliraj Karthikbabu

---

## [Editor Report · Acceptance letter]

11 Dec 2023

PONE-D-22-27895R2 

A novel motorized office chair causes low-amplitude spinal movements and activates trunk muscles: a cross-over trial 

Dear Dr. Schäfer:

I'm pleased to inform you that your manuscript has been deemed suitable for publication in PLOS ONE. Congratulations! Your manuscript is now with our production department. 

Kind regards, 

on behalf of

Dr. Monika Błaszczyszyn 

Academic Editor

PLOS ONE